# Multi-purpose cash transfers and health among vulnerable Syrian refugees in Jordan: A prospective cohort study

**Emily Lyles**[1], **Stephen Chua**[2], **Yasmeen Barham**[2], **Dina Jardenah**[3], **Antonio Trujillo**[1], **Paul Spiegel**[1], **Ann Burton**[4], **Shannon Doocy**[1]*

**1** Department of International Health, Johns Hopkins Bloomberg School of Public Health, Baltimore, Maryland, United States of America, **2** Medair, Amman, Jordan, **3** United Nations High Commissioner for Refugees, Amman, Jordan, **4** United Nations High Commissioner for Refugees, Geneva, Switzerland

* doocy1@jhu.edu

**Data Availability Statement:** The datasets supporting the conclusions of this article are available in the Humanitarian Data Exchange, and [upon acceptance] can be accessed at https://data.

## Abstract

Cash assistance has rapidly expanded in the Syrian refugee response in Jordan and global humanitarian programming, yet little is known about the effect of multipurpose cash transfers (MPC) on health in humanitarian contexts. A prospective cohort study was conducted from May 2018 through July 2019 to evaluate the effectiveness of MPC in improving access to healthcare and health expenditures by Syrian refugees in Jordan. Households receiving MPCs (US$113–219 monthly) were compared to control households not receiving MPCs using difference-in-difference analyses. Overall health care-seeking was consistently high (>85%). Care-seeking for child illness improved among MPCs but declined among controls with a significant adjusted difference in change of 11.1% (P<0.05). In both groups, child outpatient visits significantly increased while emergency room visits decreased. Changes in care-seeking and medication access for adult acute illness were similar between groups; however, hospital admissions decreased among MPCs, yet increased among controls (-8.3% significant difference in change; P<0.05). There were no significant differences in change in chronic illness care utilization. Health expenditures were higher among MPCs at baseline and endline; the only significant difference in health expenditure measures' changes between groups was in borrowing money to pay for health costs, which decreased among MPCs and increased among controls with an adjusted difference in change of -10.3% (P<0.05). The impacts of MPC on health were varied and significant differences were observed for few outcomes. MPC significantly improved care-seeking for child illness, reduced hospitalizations for adult acute illness, and lowered rates of borrowing to pay for health expenditures. No significant improvements in chronic health condition indicators or shifts in sector of care-seeking were associated with MPC. While MPC should not be considered as a stand-alone health intervention, findings may be positive for humanitarian response financing given the potential for investment in MPC to translate to health sector response savings.

humdata.org/dataset/mpc-for-syrian-refugee-health-in-jordan.

**Funding:** This research was funded by Elrha's Research for Health in Humanitarian Crises (R2HC) Programme under grant number 28370 received by SD. The R2HC programme is funded equally by the Wellcome Trust and the UK Government. The funders had no role in study design, data collection and analysis, decision to publish, or preparation of the manuscript. For more information, see: https://www.elrha.org/programme/research-for-health-in-humanitarian-crises.

**Competing interests:** The authors have declared that no competing interests exist.

## Introduction

Since 2011, more than 5.6 million people have fled conflict in Syria for neighboring countries throughout the region [1]. Jordan currently hosts over 650,000 Syrian refugees, more than 80% of whom live outside of refugee camps [2]. The humanitarian response in Jordan largely provides assistance to refugees outside of camps through existing Jordanian systems. Health assistance is provided at public sector facilities throughout the country for reduced rates based upon policies decided by the Government of Jordan (GoJ). GoJ health policies have shifted numerous times since the initial policy providing free care for Syrian refugees registered with the United Nations High Commissioner for Refugees (UNHCR) at Ministry of Health (MoH)/public sector facilities. This policy was revised in November 2014, when refugees were required to pay the same rates as uninsured Jordanians. In February/March 2018, the policy was modified again, requiring that refugees pay 80% of foreigner rates for health services directly to the MoH facility with no exemptions, resulting in out-of-pocket costs two to five times those incurred under the 2014–2018 policy [3]. Most recently, in March 2019 the health policy reverted back to provision of care for refugees at uninsured Jordanian rates with payment required directly at facilities and exemptions for services at MoH affiliated maternity and childhood centers [4]. Drivers of Syrian refugees not seeking or receiving necessary health services and medicines have fluctuated with many of these changes in GoJ policies, reflective of financial barriers consistently remaining the dominant barriers reported by Syrian refugees [5–7]. Far less often, those not receiving needed care/medicines have reported this stemming from limited availability of services/medicine, not knowing where to go, and long wait times, among other infrequently reported reasons not as directly impacted by the changing policies [5–7].

Aside from sector-specific responses, cash assistance has rapidly expanded in Jordan's humanitarian response and in global humanitarian programming [8, 9]. In response to persistent funding shortfalls, the Grand Bargain, an outcome of the 2016 World Humanitarian Summit, generated a series of commitments for donors and implementing agencies, among which is increasing routine use of cash assistance as compared to in-kind assistance [10]. Approximately US $5.6 billion in humanitarian assistance was disbursed through cash and voucher assistance (CVA) in 2019, double that of 2015 [11]. Notably, from 2018 to 2019, two of the largest UN humanitarian agencies, the World Food Program (WFP) and UNHCR, both substantially increased use of CVA; WFP increased provision of CVA from US$1.7 billion in 2018 to US$2.1 billion in 2019 (a 23% increase) and UNHCR scaled CVA from US$568 million to US$650 million (14% increase) [11, 12]. This increase is also reflected in funding Jordan's humanitarian response. Of the total US$10.4 million allocated through the Jordan Humanitarian Fund in 2018, US$3.9 million (38%) was apportioned to cash assistance [13].

In Jordan, UNHCR and WFP provide the largest refugee cash assistance programs. UNHCR's cash assistance was created in 2008 to assist Iraqi refugees and expanded in 2012 to provide increasing numbers of Syrian refugees with additional means for meeting their basic needs; it is now UNHCR's second largest cash assistance program [14]. Syrian refugee households' eligibility for receipt of UNHCR cash assistance in Jordan is determined through the Vulnerability Assessment Framework (VAF), which uses data collected during home visits to create vulnerability ratings to identify those most vulnerable; using this vulnerability ranking, households are prioritized for assistance targeting in line with the level of funding available [15]. The range of multi-sectoral indicators and methods used to generate vulnerability rankings minimize incentives to manipulate information provided during home visits. This targeting method is comprehensively revised every two years (including during the period in which the present study was implemented) to ensure the most vulnerable households are targeted,

leading to some existing beneficiaries no longer receiving MPC from UNHCR and others not previously receiving UNHCR MPC to begin receiving this assistance. UNHCR beneficiary lists may also change from one month to the next because of births/deaths and addition of new beneficiaries when previous beneficiaries are resettled to third countries [15]. Through UNHCR, eligible households receive unrestricted monthly transfers valued between 80–155 JOD (US$113–219) depending upon household size and specific vulnerabilities [15]. Barriers to accessing UNHCR MPCs in Jordan are tracked in UNHCR's regular post distribution monitoring, which, at the time of this study, indicated primary access challenges to include insufficient number of ATMs in certain governorates, lingering issues with the iris scan mechanism for accessing MPCs through ATMs, and limited knowledge of how to operate ATM equipment, though many of these difficulties have decreased in frequency over time [15].

WFP similarly provides cash transfers to vulnerable refugee households outside of camps through their "Choice" program. WFP historically provided food assistance to Syrian refugees in Jordan through in-kind food, paper and electronic vouchers, and at a smaller scale, cash [16]. In August 2017, with the introduction and subsequent expansion of the "Choice" program, WFP shifted to providing food assistance through unrestricted/multipurpose cash (MPC). Unlike UNHCR, cash through WFP's Choice program can be redeemed as cash or by purchasing food items at one of WFP's 200 partner shops throughout the country [17]. In June 2018, WFP's Choice program reached 186,732 beneficiaries across four governorates of Jordan and by August 2018 increased to a total of 292,226 beneficiaries across seven governorates with continued expansion during the period in which the present study was implemented [16]. Since April 2018, WFP transfers for Syrian refugees living outside of camps are valued from 23 JOD/US$32 to 15 JOD/US$21 per person per month [15].

Cash assistance, and particularly MPC, is generally more efficient than in-kind assistance and more supportive of beneficiary dignity and local economies [18]. The current evidence base comparing cash to in-kind assistance is largely concentrated in sectors outside of health, leaving a dearth of research comparing cash assistance to in-kind health assistance (e.g., in-kind assistance to health facilities) or other health financing approaches Cash assistance can, however, improve health through a range of channels including through both structural and intermediate determinants of health. Owusu-Addo et al.'s 2019 conceptual framework of cash transfers and the social determinants of health posits that cash transfers impact structural determinants such as financial poverty, education, productive capacity, and gender/women's empowerment, among several others; these impacts in turn positively influence intermediate determinants such as material (e.g., food security, housing) and psychosocial circumstances, which then impact utilization of health services and health outcomes [19]. By reducing financial barriers, cash assistance has been shown to improve health service utilization in several contexts with a clear bi-directional relationship between socioeconomic status and overall health, though Owusu-Addo note that the nature of health systems and contextual factors at national, local, and household levels can also directly influence utilization. Bailey and Hedlund's 2012 review of the impact of cash transfers on nutrition in emergency and transitional contexts lays a conceptual framework for humanitarian settings, exploring potential ways that emergency cash transfers can impact the causes of malnutrition, including through cash's impact on health and health behaviors [20]. Though not exclusive to MPC, this review indicated that cash transfers can impact health through increased household expenditures on healthcare (or indirectly through expenditure on hygiene products). While financial barriers are central to the cash—health pathway, cash may also improve health behaviors and outcomes by reducing employment demands (e.g., number of working hours) that may limit household members' abilities to utilize healthcare [20, 21]. Even when not directly applied for health expenditures, unconditional cash assistance may be spent in other areas that facilitate

beneficiaries' inclusion in "a health-promoting social group" [21]. Combined, these mechanisms potentially underpinning the effects of MPC on health may be realized most clearly through increased health care-seeking, improved treatment adherence, and mitigation of stress, nutrition, and numerous other lifestyle practices known to impact overall health [20–22].

While there is ample evidence in support of the benefits of cash assistance in development contexts, little is known about the effects of cash on sector-specific outcomes such as health in humanitarian contexts [21, 23–25]. It is unclear whether cash transfers would have measurable impacts on health due to the diversity of health needs across households and their members, and transfer amounts relative to the extent of unmet needs and household priorities for other types of expenditures (e.g., food, rent). In light of this gap, this study examined the effects of MPC on health-seeking behavior, health service utilization, and health expenditures among Syrian refugees in Jordan to provide evidence to inform use of cash transfer programs in both the current and future humanitarian responses.

## Methods

A prospective cohort study was conducted from May 2018 through July 2019 to evaluate the effectiveness of MPC provided by UNHCR to vulnerable Syrian refugee households in increasing access to health. Systematically sampled households receiving MPC from UNHCR at the start of this study (intervention group) and similarly vulnerable households not receiving MPC (control group) were followed for one year to compare health expenditures, health-seeking behavior, and health service utilization between the two groups. In an effort to understand the effect of MPC on health in multiple contexts, a parallel study was also conducted in Lebanon in the same timeframe, utilizing the same study design to examine comparable intervention in a setting with important differences (e.g., health system structures, provision of humanitarian assistance, and refugee settlement patterns) [26]. For the purposes of this study, households were defined as people who share a living space and share both meals and financial resources.

### Sampling

Due to strategic targeting of MPC assistance to vulnerable Syrian refugee households in response to funding limitations, at the time of study initiation 30% of registered refugee households (n = 27,932) received MPC from UNHCR while approximately 92% of households were classified as highly or severely vulnerable with respect to basic needs in the 2017 Jordan VAF sector vulnerability review [27]. This distribution of households with similar levels of economic vulnerability (not) receiving MPC facilitated assessment of health outcomes associated with cash assistance in two reasonably comparable groups of households. To reduce variability between comparison groups and the risk of cross-over (i.e., existing beneficiaries stopping receipt of MPC and/or those not receiving MPC becoming MPC beneficiaries) due to changes in targeting methods based upon predicted household expenditures and anticipated scale up of MPC programs, the sample was restricted to households with predicted per capita monthly expenditures between 40–50 JOD/US$56–70.

Sample size calculations were based on the primary aim of comparing households receiving MPC to similar households not receiving MPC. Most outcome measures of interest (e.g., care-seeking or having out-of-pocket health expenditures) can be expressed as a proportion, thus, in the absence of comparable studies at the time of study design, calculations assumed the most conservative proportion of 50%, power = 0.80, a minimum detectable difference of ≥10%, and were two-sided. Based on these assumptions, a minimum required sample size of

770 households (385 per group) was identified; this was increased to a minimum planned sample of 1,000 households to allow for loss to follow-up of ≤30%. The sample (n = 1,000) was allocated by governorate proportionally to the location of UNHCR MPC beneficiaries with similar numbers in the intervention and control groups. Lists of MPC recipients and non-MPC recipients were ordered by estimated per capita expenditure and systematically sampled. Sampling lists included additional households to the projected sample to allow for replacement sampling when households were unreachable, declined to participate, or determined to be ineligible during screening questions ahead of enrollment interviews. Syrian refugees (87%) and as a result, MPC beneficiaries and the study sample, are concentrated in four governorates of Jordan (Amman, Irbid, Mafraq, and Zarqa) with relatively few in the remaining eight governorates.

UNHCR revises targeting for MPC recipients every two years to reflect changes in households' financial situations and ensure those currently considered most vulnerable are selected. Consequently, over the course of the study period, some existing beneficiaries stopped receiving MPC (n = 121; 24.2%) while others not previously receiving MPC were added to the MPC beneficiary list (n = 51; 10.2%). To maximize statistical power in light of sample changes, participants receiving MPC from UNHCR at endline (i.e., those who received MPC for the entire study period and those who began receiving MPC at the study mid-point) were analyzed as MPC beneficiary households (intervention group) while the control group included only those not receiving MPC through the entire study period. Additional detail on change in intervention receipt is presented in S1 File.

## Study implementation

UNHCR lists of registered refugee households, including household names, phone number, district of residence, vulnerability category, and receipt of cash assistance were used for recruitment. This information was used only by the study coordinator to identify prospective participants. Interviewers that conducted recruitment received training on privacy and confidentiality and were provided only with names and phone numbers to reduce the risk of sensitive information (such as vulnerability status) being shared. Records of all participant phone numbers and identifiable information were destroyed immediately after use for the study. Sampled households were contacted by phone and invited to participate; households were asked a series of questions to confirm eligibility before the enrollment interview. In order to participate, households were required to be Syrian refugees that are currently registered with UNCHR, classified as vulnerable or severely vulnerable per UNHCR, and registered with UNHCR as residing outside of a camp setting. All consenting eligible households were enrolled in the study until the target sample size (n = 1,000) was reached. Respondents were household heads or principal applicants on the sampled UNHCR registration case when possible; if that individual did not regularly reside with the household or could not be reached, another adult member of the sampled household (who thus also met eligibility criteria) was permitted to complete interviews. If a sampled household could not be reached by phone after three attempts, they were deemed ineligible to participate in the study.

Verbal informed consent was required prior to enrollment interviews and an abbreviated oral consent for continued participation was used prior to endline interviews. All enrolled households completed phone interviews at enrollment (May-July 2018) and endline one-year following enrollment (May-July 2019). Phone interviews have been used by UNHCR and academic institutions for health research in Jordan and throughout the region with acceptable response rates and are regarded as an appropriate medium for data collection [5, 28, 29]. In this study, phone interviews were preferable for logistical feasibility, cost efficiency, and to

better protect participant privacy given the potential risks associated with others likely learning of their participation if interviews were in-person. Interviews lasted between 30–50 minutes and used a structured questionnaire (see S2 File) focused on household demographic and socioeconomic characteristics, receipt of humanitarian assistance, and household health seeking behavior, health service utilization, and health expenditures for child, adult acute, and chronic illnesses. Enumerators mostly had prior data collection experience and received two days of classroom training on data collection tools, mobile data collection platform, interview techniques, and basic principles of human subjects' protections followed by additional supervised practice interviews. Data were collected on Android tablets using the Magpi mobile data platform by DataDyne LLC (Washington, DC). Data collection was supervised by a local study coordinator with daily checks of uploaded interviews for quality and completeness to promptly address concerns.

### Data analysis

Data analysis was performed using Stata 13 (College Station, TX). Differences between study groups (i.e., MPC vs. non-MPC beneficiaries) in descriptive analyses were examined using chi-square and t-test methods for binary/categorical and continuous variables, respectively. Regression models were used to evaluate the effects of MPC on health outcomes, both unadjusted and controlling for differences in household characteristics. Covariates for all adjusted models were selected *a priori* to include characteristics known or suspected to be associated with intervention receipt and outcomes of interest. Linear probability models were used to estimate differences in binary outcomes between study groups from baseline to endline with main terms for study group, time period, and the interaction between study group and time period. Log-linear models were similarly used to estimate differences in continuous outcomes; log transformation was required for health expenditure outcomes due to their skewed distribution. Coefficients for the interaction of study group and time period represent the estimated difference in change comparing MPC beneficiaries to non-beneficiaries (i.e., the difference-in-difference/"MPC effect"). Effect sizes associated with receipt of MPC were also calculated by dividing difference-in-difference (DiD) by the overall mean for each outcome. All models utilized cluster-robust standard errors with clustering defined at the household level, allowing for correlation between observations for each household.

Financial indicators are presented in U.S. Dollars (US$) using an exchange rate of 1.41 JOD/US$1 [30]. All monetary variables were assessed for outliers using visual inspection and individual consideration of points falling three or more standard deviations from the mean. Outliers believed to be the result of misreporting or entry errors were corrected or removed from the data set. Other outliers were checked with field teams for accuracy and corrected as needed. Preliminary analysis and findings were discussed by all the collaborating organizations prior to finalization of results to ensure their accuracy and the best possible interpretation of findings within the Jordanian context.

The research was approved by the Johns Hopkins Bloomberg School of Public Health Institutional Review Board and the Ministry of Planning and International Cooperation of Jordan prior to the start of the study. Additional information regarding the ethical, cultural, and scientific considerations specific to inclusivity in global research is included in the S1 Checklist.

## Results

### Study population characteristics

A total of 998 households were enrolled in the study, of which 885 (88.7%) were followed for one year and completed endline interviews. During the study period, WFP scaled coverage of

their Choice program, offering beneficiaries the option of MPC or e-vouchers and a small number of households changed beneficiary status for UNHCR MPC. Those receiving UNHCR MPC at enrollment who stopped receiving MPC during follow-up (n = 121) were excluded from the analysis. Final analyses compared participants receiving MPC from UNHCR at endline (i.e., "MPC households"/intervention group, n = 429) with control households comprising participants not receiving MPC for the entire study period (n = 448). Retention was 96% among MPC households and 87% among controls; reasons for loss to follow-up and baseline characteristics of participants lost to follow-up are provided in S1 File. The enrolled sample and the final analyzed sample are presented in Table 1.

Principal applicants on the sampled UNHCR case served as respondents in 80% and 83% of households at baseline and endline, respectively. Respondents were immediate family members of the principal applicant (e.g., husband/wife, son/daughter, mother/father) in 19% of households at baseline and 16% at endline.

Characteristics of the principal applicant, household composition, and living conditions are summarized in Table 2. At both baseline and endline, principal applicants in MPC households were more commonly female, significantly older, and comparably less educated than in control households. Control households had significantly larger average household size, though smaller dependency ratios. They were also more likely to have children, but relative to MPC households, significantly fewer control households had members 60 years or older, with a disability, or needing daily living support. At baseline, living conditions also significantly differed with control households being more likely to live in an entire apartment/house and having a higher ratio of household members to sleeping rooms in the residence.

Household economic characteristics (Table 3) differed significantly at baseline in terms of both income and expenditure, with MPC recipients reporting significantly lower incomes and expenditures. Mean incomes and expenditures decreased from baseline to endline in both groups, though only income differed significantly between groups at endline. Significant differences in receipt of any regular cash assistance were observed at baseline and endline both in the proportion of households receiving assistance and total amounts received. Notably, the proportion of households receiving WFP assistance increased in both groups over the study period and WFP transitioned similarly large proportions of households in both groups from e-vouchers to "Choice"/MPC.

### Health service utilization

Care-seeking was assessed for the most recent household member illness (within the past six months) believed to be severe enough to warrant medical care; results are reported for children's illnesses and for acute and chronic illnesses among adults. Baseline and endline

**Table 1. Enrolled sample and analyzed sample by region.**

| Region | UNHCR MPC Recipients by Region at Enrollment | Enrolled Sample (n = 998) | | | | Analyzed Sample (n = 877) | | | |
|---|---|---|---|---|---|---|---|---|---|
| | | Intervention (n = 499) | | Control (n = 499) | | Intervention (n = 429) | | Control (n = 448) | |
| | | N | % | N | % | N | % | N | % |
| Amman | 29.8% | 171 | 34.3% | 173 | 34.7% | 144 | 33.6% | 159 | 35.5% |
| Irbid | 27.4% | 126 | 25.3% | 126 | 25.3% | 101 | 23.5% | 112 | 25.0% |
| Mafraq | 18.8% | 86 | 17.2% | 88 | 17.6% | 86 | 20.0% | 72 | 16.1% |
| Zarqa | 10.6% | 51 | 10.2% | 49 | 9.8% | 41 | 9.6% | 45 | 10.0% |
| Rest of Jordan [a] | 13.3% | 65 | 13.0% | 63 | 12.6% | 57 | 13.3% | 60 | 13.4% |

[a] Rest of Jordan includes Ajloun, Aqaba, Balqa, Jerash, Karak, Maan, Madaba, and Tafiela governorates

**Table 2. Sample demographic characteristics and living conditions at baseline and endline.**

| | | BASELINE | | | ENDLINE | | |
|---|---|---|---|---|---|---|---|
| | | MPC HHs (N = 429) | Control HHs (N = 448) | P value | MPC HHs (N = 411) | Control HHs (N = 391) | P value |
| **Principal Applicant/Household Head Characteristics** | | | | | | | |
| Female sex | | 257 (59.9%) | 111 (24.8%) | *<0.001* | 245 (59.6%) | 100 (25.6%) | *<0.001* |
| Age (mean years) | | 57.1 (15.4) | 38.0 (13.5) | *<0.001* | 57.4 (15.7) | 38.9 (13.8) | *<0.001* |
| Highest level of education | None | 152 (35.6%) | 51 (11.5%) | *<0.001* | 140 (34.1%) | 43 (11.0%) | *<0.001* |
| | Primary school | 91 (37.7%) | 202 (45.4%) | | 170 (41.5%) | 210 (53.7%) | |
| | Preparatory school | 22 (12.9%) | 110 (24.7%) | | 47 (11.5%) | 71 (18.2%) | |
| | Secondary school | 7 (13.8%) | 82 (18.4%) | | 53 (12.9%) | 67 (17.1%) | |
| Marital status | Married | 193 (45.0%) | 361 (80.6%) | *<0.001* | 188 (45.7%) | 313 (80.1%) | *<0.001* |
| | Widowed | 133 (31.0%) | 22 (4.9%) | | 140 (34.1%) | 22 (5.6%) | |
| | Never married / Divorced | 43 (10.0%) | 43 (9.6%) | | 47 (11.4%) | 32 (8.2%) | |
| **Household Demographic Characteristics** | | | | | | | |
| Household size (mean) | | 4.2 (2.3) | 4.5 (2.1) | **0.028** | 4.1 (2.3) | 4.6 (1.8) | **0.001** |
| Dependency ratio [a] (mean) | | 1.4 (1.0) | 1.0 (0.8) | *<0.001* | 1.3 (1.0) | 1.1 (0.9) | **0.005** |
| Multiple UNHCR registration cases (%) | | 192 (44.8%) | 134 (29.9%) | *<0.001* | 142 (34.5%) | 94 (24.0%) | **0.001** |
| Vulnerable members (%) | | | | | | | |
| | Child(ren) <5 yrs | 132 (30.8%) | 281 (62.7%) | *<0.001* | 107 (26.0%) | 250 (63.9%) | *<0.001* |
| | Child(ren) ≤ 17 yrs | 260 (60.6%) | 366 (81.7%) | *<0.001* | 240 (58.4%) | 319 (81.6%) | *<0.001* |
| | Older adult(s) (>60 yrs) | 267 (62.2%) | 81 (18.1%) | *<0.001* | 259 (63.0%) | 75 (19.2%) | *<0.001* |
| Member with a chronic health condition | | 352 (82.1%) | 229 (51.1%) | *<0.001* | 338 (82.2%) | 192 (49.1%) | *<0.001* |
| Member w/ disability or that needs daily support | | 151 (35.2%) | 78 (17.4%) | *<0.001* | 116 (28.2%) | 46 (11.8%) | *<0.001* |
| **Living Conditions** | | | | | | | |
| Residence type | Apartment or house | 346 (80.7%) | 383 (85.5%) | **0.008** | 367 (89.3%) | 352 (90.0%) | 0.217 |
| | Single room | 43 (10.0%) | 30 (6.7%) | | 14 (3.4%) | 8 (2.0%) | |
| | Temporary shelter [b] | 21 (4.9%) | 29 (6.5%) | | 22 (5.4%) | 28 (7.2%) | |
| | Other [c] | 19 (4.4%) | 6 (1.3%) | | 8 (1.9%) | 3 (0.8%) | |
| Residence arrangement | Rented | 390 (90.9%) | 411 (91.7%) | 0.657 | 384 (93.4%) | 374 (95.7%) | 0.326 |
| | Hosted for free / rent paid by NGO/charity | 21 (4.9%) | 15 (3.3%) | | 14 (3.4%) | 8 (2.0%) | |
| | Owned | 15 (3.5%) | 19 (4.2%) | | 13 (3.2%) | 8 (2.0%) | |
| Crowding (mean # people/sleeping room) | | 4.5 (1.9) | 3.6 (1.7) | *<0.001* | 4.3 (2.0) | 4.1 (1.9) | 0.228 |

Presented as N (%) or mean (standard deviation). Bold italic indicates statistically significant (P < 0.001) findings; bold indicates statistically significant (P < 0.05) findings; italic indicates statistically significant (P < 0.10) findings.

[a] Number of dependents divided by number of working age adults

[b] includes tent, prefab unit, collective center

[c] includes unfinished building, construction site, factory, or warehouse

descriptive comparisons of both groups are presented in S1 Table and in Fig 1; unadjusted and adjusted individual group change and differences in change between groups are provided in Table 4 and Fig 2.

Reasons for needing medical care for a child's illness were similar between MPC and control households and most commonly included respiratory infections, fever, diarrhea, and asthma (Fig 1). Care-seeking rates for childhood illness were significantly greater at baseline among controls than MPC recipients (87.4% vs. 77.0%, P = 0.014), but decreased, though not significantly, by endline with a significant 11.1% adjusted difference in change between groups (CI: 0.8–21.3%; P = 0.035; effect size: 13.3%). Small sample sizes for those not seeking care

**Table 3. Household economy and receipt of humanitarian assistance.**

| | | BASELINE | | | ENDLINE | | |
|---|---|---|---|---|---|---|---|
| | | MPC HHs (N = 429) | Control HHs (N = 448) | P value | MPC HHs (N = 411) | Control HHs (N = 391) | P value |
| **Household Income and Expenditures (past month; mean US$ [a])** | | | | | | | |
| Income (excluding humanitarian assistance) | | 305 (566.8) | 445 (598.9) | *<0.001* | 193 (423.9) | 356 (277.7) | *<0.001* |
| Total expenditures | | 471 (377.7) | 556 (552.9) | **0.008** | 466 (327.3) | 474 (244.8) | 0.710 |
| **Total Humanitarian Assistance (past month) [b]** | | | | | | | |
| **Any regular cash transfer (% of HHs)** | | 413 (96.3%) | 387 (86.4%) | *<0.001* | 411 (100.0%) | 346 (88.5%) | *<0.001* |
| Amount received (mean US$ [a] per HH) | | 215 (121.3) | 101 (67.2) | *<0.001* | 239 (97.9) | 99 (77.0) | *<0.001* |
| Amount received (mean US$ [a] per HH member) | | 63 (36.0) | 23 (14.6) | *<0.001* | 72 (33.8) | 22 (18.2) | *<0.001* |
| **In-kind assistance (past 3 months)[b]** | | 22 (5.1%) | 27 (6.0%) | 0.703 | 61 (14.8%) | 52 (13.3%) | 0.490 |
| **WFP Food Assistance (past month)** | | | | | | | |
| Current WFP recipients | | 397 (92.5%) | 387 (86.4%) | **0.011** | 400 (97.3%) | 343 (87.7%) | *<0.001* |
| Amount received (mean US$ [a] per HH) | | 89 (63.6) | 90 (48.9) | 0.785 | 87 (59.5) | 99 (50.1) | **0.003** |
| Amount received (mean US$[a]per HH member) | | 22 (8.4) | 20 (7.1) | **0.010** | 22 (7.4) | 21 (8.8) | 0.697 |
| Transfer modality | E-Voucher | 304 (76.6%) | 300 (77.5%) | 0.753 | 118 (29.5%) | 99 (28.9%) | 0.849 |
| | Choice | 93 (23.4%) | 87 (22.5%) | | 282 (70.5%) | 244 (71.1%) | |
| **Asset Sales and Borrowing** | | | | | | | |
| Sold assets in past 3 months (%) | | 76 (17.7%) | 119 (26.6%) | **0.002** | 90 (21.9%) | 127 (32.5%) | **0.003** |
| Borrowed money in past 3 months (%) | | 271 (63.2%) | 310 (69.2%) | **0.018** | 302 (73.5%) | 328 (83.9%) | **0.002** |
| Current debt | Any Debt | 304 (76.6%) | 345 (82.7%) | **0.029** | 321 (84.0%) | 346 (93.5%) | *<0.001* |
| Amount of debt (among those w/ debt; mean US$ [a]) | | 633 (1201.8) | 701 (1735.0) | 0.519 | 1340 (7427.6) | 1182 (1711.0) | 0.691 |

Presented as N (%) or mean (SD). Bold italic indicates statistically significant (P < 0.001) findings; bold indicates statistically significant (P < 0.05) findings; italic indicates statistically significant (P < 0.10) findings.

[a] Exchange rate: 1 JOD = 1.41 US$; [b] includes UNHCR, WFP, and regular monthly assistance from other less common sources

[b] Includes direct distribution of or assistance for accommodation, shelter materials/repairs; utilities (fuel, electricity, water); household items; clothing; food; medication; health services; education; business/livelihood inputs

prohibited robust analysis of change; however, among households that did not seek needed care, cost was the most commonly reported reason. The proportion of households that sought care who reported not receiving all recommended care due to cost decreased from baseline to endline in both groups; however, as with nearly all child health utilization outcomes, the difference in change between groups was not statistically significant in adjusted analyses (P = 0.148).

While most children received care as outpatient visits, at baseline 26% of MPC group children and 20% of control children were treated at the emergency room (ER) and 3% and 2% of children, respectively, had a hospital admission the last time care was sought. At endline, both groups reported decreases in emergency room visits for child illness and increases in outpatient visits (P<0.001 for all within-group changes); however, no significant difference in adjusted facility utilization change was observed between groups (outpatient P = 0.097; ER P = 0.091). Both MPC recipients and controls predominantly sought care for childhood illness in the private sector and, while private sector care-seeking decreased in both groups at endline, change was small, not statistically significant (adjusted MPC change P = 0.301; control change

**Childhood Illness**

The most commonly reported reasons for needing medical care for children included:

- Respiratory infections (45% baseline / 48% endline)
- Fever (17% baseline / 18% endline)
- Diarrhea (6% baseline / 8% endline)
- Asthma (3% baseline / 5% endline)

Reasons for needing care were similar between MPC and control households at baseline (P=0.062) and endline (P=0.249).

**Adult Acute Illness**

Infection was the most common reason for adults needing care at both baseline (45%) and endline (72%) followed by dental (7% baseline / 8% endline) and gynecological problems (11% baseline / 3% endline). Reasons for care-seeking were similar between MPC and control households at baseline (P=0.488) but differed significantly at endline (P=0.001) where MPC households reported infection more frequently than control households, who reported dental and gynecological problems as reason for needing care more frequently (11.8% vs 5.6% and 5.3% vs 1.3%, respectively).

**Adult Chronic Illness**

The proportion of households with member(s) with chronic health conditions was significantly higher among MPC beneficiaries compared to controls at baseline (82% vs 51%) and endline (82% vs 49%). Hypertension and diabetes adult prevalence rates were 21.6% and 13.6%, respectively, at baseline. Among household members with chronic condition(s), the most frequently reported were hypertension (51% baseline/56% endline), diabetes (33% baseline/34% endline), arthritis (26% baseline/22% endline), and cardiovascular disease (24% baseline/22% endline).

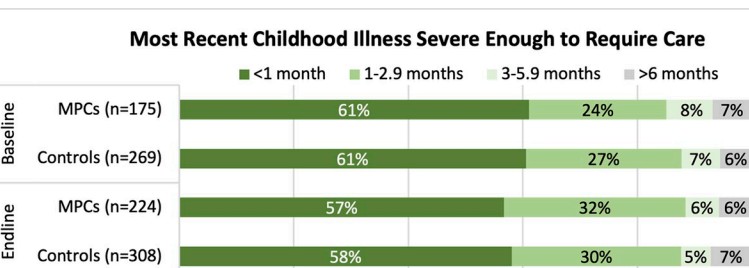

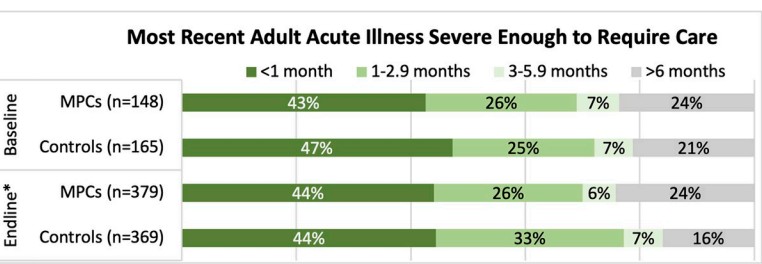

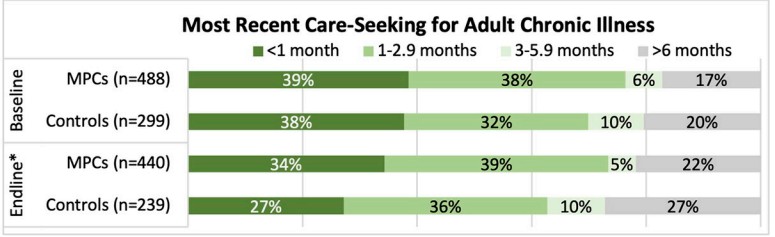

*indicates statistically significant difference between MPCs and controls

**Fig 1. Reasons and timeframes for care-seeking.**

P = 0.508), and did not significantly differ between groups (P = 0.686). Access to medication for childhood illness was high, with more than 97% of households in both MPC and control groups obtaining prescribed medications at both time periods. At endline, increases in obtaining medication were observed in both groups, but the increase was not significantly different among controls compared to MPC recipients (P = 0.333).

Compared to MPC recipients, control households reported needing care for an adult with acute illness more recently (Fig 1), though these differences were significant only at endline (baseline P = 0.867; endline P = 0.021). Reasons for adult care-seeking were similar between MPC and control households at baseline (P = 0.488) but differed significantly at endline (P = 0.001) when MPC households reported infection more commonly than controls, who more frequently cited needing care for dental and gynecological problems (Fig 1). Care-seeking rates were significantly higher in control households than MPC households at baseline (73.5% and 68.3%, respectively; P = 0.031) and small but statistically non-significant decreases in care-seeking were observed in both MPC (P = 0.415) and control (P = 0.857) households (adjusted DiD P = 0.618). Cost remained the primary reason for not seeking care; and while smaller proportions of households in both groups reported not receiving all needed care due to cost at endline compared to baseline, change was only significant among MPC recipients (-10.3%, CI: -20.5,-0.1; P = 0.047) and did not significantly differ from controls (adjusted DiD P = 0.328).

Although most adults received care for acute illness through outpatient visits, significantly more MPC households than controls reported emergency room visits (20.7% and 11.0%,

**Table 4. Change in health care-seeking and medicines for household member illness.**

| | | | UNADJUSTED CHANGE OVER TIME | | | | | | ADJUSTED CHANGE OVER TIME [a] | | | | | | Effect Size | R² |
|---|---|---|---|---|---|---|---|---|---|---|---|---|---|---|---|---|
| | | | MPC HHs | | Control HHs | | DiD | | MPC HHs | | Control HHs | | DiD | | | |
| | | | % | (95% CI) | % | (95% CI) | % | (95% CI) | % | (95% CI) | % | (95% CI) | % | (95% CI) | | |
| **Most Recent Childhood Illness** | | | | | | | | | | | | | | | | |
| Sought and received medical care | | | 5.9% | (-2.3,14.1) | -3.1% | (-8.7,2.5) | 9.0% | *(-0.9,18.9)* | 9.3% | (0.7,17.9) | -1.8% | (-7.8,4.2) | **11.1%** | **(0.8,21.3)** | 13.3% | 0.038 |
| Able to obtain prescribed meds | | | **3.4%** | **(0.1,6.7)** | 0.7% | (-1.7,3.0) | 2.7% | (-1.3,6.7) | 2.5% | (-0.7,5.7) | 0.3% | (-2.2,2.8) | 2.2% | (-2.2,6.5) | 2.2% | 0.036 |
| Outpatient visit | | | *22.2%* | *(13.6,30.8)* | *16.2%* | *(10.1,22.4)* | 6.0% | (-4.6,16.6) | 23.0% | (14.0,32.1) | 13.7% | (7.1,20.4) | 9.3% | *(-1.7,20.2)* | 10.9% | 0.097 |
| Emergency room visit | | | *-21.9%* | *(-30.1,-13.6)* | *-15.7%* | *(-21.5,-9.9)* | -6.2% | (-16.3,3.9) | *-22.9%* | *(-31.5,-14.3)* | *-13.9%* | *(-20.2,-7.6)* | -9.0% | *(-19.4,1.4)* | -70.7% | 0.101 |
| Hospital admission | | | -0.3% | (-4.3,3.6) | -0.6% | (-3.1,1.9) | 0.2% | (-4.4,4.9) | -0.1% | (-4.0,3.8) | 0.2% | (-2.5,2.9) | -0.3% | (-5.2,4.6) | -13.0% | 0.017 |
| Care Facility Sector | Public | | 3.9% | (-7.0,14.8) | 1.0% | (-7.6,9.6) | 2.9% | (-11.0,16.8) | 4.5% | (-7.5,16.6) | 2.1% | (-7.0,11.1) | 2.5% | (-12.4,17.4) | 10.2% | 0.024 |
| | Private | | -4.5% | (-16.1,7.1) | -2.0% | (-11.2,7.2) | -2.5% | (-17.3,12.3) | -6.4% | (-18.6,5.8) | -3.3% | (-12.9,6.4) | -3.2% | (-18.6,12.2) | -4.9% | 0.059 |
| | Charity | | 0.6% | (-7.3,8.6) | 1.1% | (-4.3,6.4) | -0.4% | (-10.0,9.2) | 1.9% | (-6.2,10.0) | 1.2% | (-4.6,6.9) | 0.7% | (-9.4,10.8) | 6.2% | 0.051 |
| Medical care not sought b/c of cost | | | *7.9%* | *(-0.9,16.7)* | 4.2% | (-10.3,18.7) | 3.7% | (-13.3,20.7) | 5.7% | (-3.3,14.8) | 8.5% | (-7.3,24.3) | -2.8% | (-22.5,17.0) | -3.0% | 0.109 |
| All needed care not received due to cost | | | **-11.3%** | **(-18.3,-4.2)** | -4.4% | *(-9.5,0.6)* | -6.8% | (-15.5,1.8) | **-8.3%** | **(-16.0,-0.7)** | -1.8% | (-7.2,3.6) | -6.6% | (-15.4,2.3) | -68.3% | 0.063 |
| **Most Recent Adult Acute Illness** | | | | | | | | | | | | | | | | |
| Sought and received medical care | | | -4.4% | (-14.0,5.2) | -5.4% | (-14.4,3.6) | 1.0% | (-12.2,14.2) | -4.7% | (-15.9,6.6) | -0.9% | (-10.8,9.0) | -3.8% | (-18.6,11.0) | -5.6% | 0.029 |
| Able to obtain prescribed meds | | | 4.2% | (-1.1,9.6) | **4.4%** | **(0.0,8.7)** | -0.1% | (-7.0,6.8) | *5.4%* | *(-0.4,11.2)* | 3.2% | *(-0.5,6.9)* | 2.2% | (-3.7,8.1) | 2.2% | 0.056 |
| Outpatient visit | | | *25.5%* | *(15.8,35.2)* | 5.1% | (-1.9,12.1) | *20.4%* | *(8.4,32.3)* | *20.9%* | *(9.8,32.0)* | 5.0% | (-2.8,12.9) | **15.8%** | **(2.5,29.2)** | 17.4% | 0.103 |
| Emergency room visit | | | *-18.1%* | *(-26.8,-9.5)* | -5.1% | (-12.1,1.9) | **-13.1%** | **(-24.2,-1.9)** | **-12.7%** | **(-22.2,-3.2)** | -5.1% | (-12.9,2.7) | -7.6% | (-19.5,4.4) | -93.4% | 0.082 |
| Hospital admission | | | **-7.3%** | **(-13.0,-1.6)** | 0.0% | – | **-7.3%** | **(-13.0,-1.6)** | **-8.2%** | **(-15.0,-1.3)** | 0.1% | (-0.6,0.8) | **-8.3%** | **(-15.2,-1.3)** | -898.2% | 0.085 |
| Care Facility Sector | Public | | 0.3% | (-13.1,13.7) | 6.2% | (-4.5,16.9) | -5.9% | (-23.1,11.2) | 1.0% | (-12.9,14.9) | 4.9% | (-6.6,16.4) | -3.9% | (-21.7,13.8) | -17.3% | 0.069 |
| | Private | | -4.8% | (-19.5,10.0) | **-12.8%** | **(-25.1,-0.5)** | 8.0% | (-11.2,27.2) | -4.5% | (-20.7,11.7) | *-12.9%* | *(-26.3,0.4)* | 8.4% | (-12.6,29.5) | 13.9% | 0.102 |
| | Charity | | 4.5% | (-6.8,15.8) | 6.6% | (-1.8,15.1) | -2.1% | (-16.2,12.0) | 3.5% | (-8.7,15.7) | 8.0% | *(-1.5,17.6)* | -4.5% | (-20.4,11.3) | -27.5% | 0.072 |
| Medical care not sought b/c of cost | | | 10.3% | (-2.4,23.0) | **15.6%** | **(2.0,29.2)** | -5.3% | (-23.9,13.3) | 11.4% | (-3.3,26.1) | **14.8%** | **(0.8,28.8)** | -3.4% | (-23.5,16.8) | -3.7% | 0.084 |
| All needed care not received due to cost | | | *-17.7%* | *(-27.5,-7.8)* | -6.9% | (-15.2,1.4) | -10.7% | (-23.6,2.2) | **-10.3%** | **(-20.5,-0.1)** | -3.7% | (-12.5,5.0) | -6.6% | (-19.8,6.6) | -52.9% | 0.068 |
| **Adult Chronic Illness** | | | | | | | | | | | | | | | | |
| Sought and received medical care | | | **3.3%** | **(0.6,6.0)** | **5.2%** | **(1.3,9.0)** | -1.8% | (-6.5,2.8) | 2.4% | (-1.5,6.3) | **5.2%** | **(1.2,9.3)** | -2.8% | (-8.1,2.4) | -3.0% | 0.014 |
| Ever faced difficulties obtaining meds | | | **-10.9%** | **(-17.3,-4.4)** | **-10.4%** | **(-19.0,-1.8)** | -0.5% | (-11.3,10.3) | **-14.0%** | **(-21.9,-6.1)** | **-10.7%** | **(-20.1,-1.2)** | -3.3% | (-15.0,8.3) | -5.2% | 0.037 |
| General practitioner visit(s) | | | 2.7% | (-4.4,9.7) | 6.0% | (-4.5,16.4) | -3.3% | (-15.9,9.3) | 4.2% | (-4.3,12.6) | 5.2% | (-5.9,16.4) | -1.1% | (-14.6,12.5) | -1.7% | 0.009 |
| Specialist visit(s) | | | -2.7% | (-9.9,4.4) | *-9.2%* | *(-18.4,0.1)* | 6.4% | (-5.3,18.1) | -1.1% | (-9.7,7.5) | **-10.6%** | **(-20.6,-0.6)** | 9.5% | (-3.0,22.0) | 16.9% | 0.033 |
| Hospital visit(s) | | | **-7.6%** | **(-14.7,-0.4)** | **-14.1%** | **(-22.5,-5.6)** | 6.5% | (-4.6,17.6) | -2.6% | (-11.1,5.9) | **-11.3%** | **(-20.3,-2.4)** | 8.7% | (-3.0,20.4) | 23.5% | 0.033 |
| Care Facility Sector | Public | | -2.6% | (-8.2,3.0) | **-8.9%** | **(-17.1,-0.7)** | 6.4% | (-3.6,16.3) | -1.4% | (-8.3,5.6) | **-9.6%** | **(-18.2,-1.1)** | 8.3% | (-2.3,18.9) | 37.3% | 0.028 |
| | Private | | -3.1% | (-9.3,3.2) | 5.6% | (-3.0,14.1) | -8.6% | (-19.2,2.0) | -2.3% | (-10.4,5.7) | 5.8% | (-3.7,15.2) | -8.1% | (-19.7,3.5) | -18.7% | 0.024 |
| | Charity | | 5.6% | (-0.2,11.5) | 3.4% | (-4.2,10.9) | 2.3% | (-7.3,11.8) | 3.7% | (-3.5,10.9) | 3.9% | (-4.2,11.9) | -0.2% | (-10.7,10.4) | -0.5% | 0.048 |
| Medical care not sought b/c of cost | | | 2.5% | (-38.4,43.4) | **29.4%** | **(6.8,52.0)** | -26.9% | (-73.6,19.8) | -11.4% | (-59.5,36.7) | 25.6% | (-38.2,89.5) | -37.0% | (-117.5,43.5) | -57.2% | 0.378 |
| All needed care not received due to cost | | | **-10.6%** | **(-17.1,-4.1)** | -3.7% | (-11.3,4.0) | -6.9% | (-17.0,3.1) | **-9.6%** | **(-17.4,-1.9)** | -0.4% | (-8.6,7.8) | -9.3% | *(-20.0,1.5)* | -35.4% | 0.029 |
| Cannot afford medication | | | *-19.5%* | *(-25.9,-13.0)* | **-15.1%** | **(-23.5,-6.7)** | -4.3% | (-15.0,6.3) | *-20.7%* | *(-28.6,-12.9)* | **-15.8%** | **(-25.1,-6.6)** | -4.9% | (-16.5,6.7) | -9.2% | 0.059 |

DiD (difference-in-difference) represents the difference in change between MPC & control groups; effect size indicates the magnitude of this difference (DiD estimate divided by overall mean).

Bold italic indicates statistically significant (P < 0.001) findings; bold indicates statistically significant (P < 0.05) findings; italic indicates statistically significant (P < 0.10) findings.

[a] Adjusted analyses controlled for principal applicant sex, age, education level, and marital status; household size and composition including the presence of household members in need of daily support, with chronic conditions, and children under five years old; dependency ratio; total household expenditure in the prior month; and receipt of humanitarian assistance, specifically total value of cash assistance received in the prior month, and current WFP beneficiary status/transfer modality (voucher or Choice).

respectively) and inpatient admissions (7.3% and 0.0%, respectively) at baseline. Care-seeking location for acute adult illness was similar between the two groups at endline and changed over time; in adjusted models comparing change in the two groups, outpatient care-seeking increased by 15.8% (CI: 2.5,29.2; P = 0.020) more in the MPC group compared to controls (effect size: 17.4%). Hospital admissions were reported by 7.3% of MPC recipient adult acute illness care-seekers but no controls at baseline nor in either group at endline, leading to an adjusted difference in change of -8.3% (CI: -15.2,-1.3; P = 0.021). At baseline, a significantly larger proportion of control households sought care in the private sector, while more MPC households sought care at public sector or charity facilities; sector utilization was similar between the two groups at endline. In adjusted analyses, private sector utilization decreased among controls and MPC recipients; both groups increased public and charity sector utilization over the study period; however, none of these changes significantly differed between groups (private sector P = 0.431; public P = 0.633; charity P = 0.576). Like childhood illnesses, access to medication for adult acute illness was very high, with more than 94% of households in both groups reporting they were able to obtain prescribed medications; this did not significantly change in either group (adjusted DiD P = 0.467).

The proportion of households with member(s) with chronic health conditions was significantly higher among MPC beneficiaries compared to controls at baseline (82% vs 51%) and endline (82% vs 49%). Among household members with chronic condition(s), the most frequently reported were hypertension, diabetes, arthritis, and cardiovascular disease (Fig 1). MPC and control households did not significantly differ at either baseline or endline in nearly all care utilization outcomes for adult chronic illnesses. Households receiving MPC reported seeking care for adult chronic illnesses more recently than controls (Fig 1), but this difference was significant only at endline (P = 0.010). Both study groups reported similarly high care-seeking rates at baseline and endline with more than 93% of individuals with a chronic condition having received care for their condition in Jordan. Similar proportions of MPC and control participants reported visits to a general practitioner and/or specialist in the preceding six months. Hospital visits were also similarly reported by both groups at baseline and while both groups decreased during follow-up, a significantly (P = 0.035) greater proportion of MPC participants (34.7%) reported hospital visits at endline compared to controls (26.0%). No statistically significant differences in adjusted utilization change were observed between groups. As with child and adult acute illness, sector utilization for adult chronic illnesses were similar between MPC and control groups and, except for decreased public sector utilization among controls (-9.6%, CI: -18.2,-1.1; P = 0.027), did not significantly change or differ between groups during follow-up. Significantly fewer MPC recipients reported not receiving needed care/services because of cost at endline (adjusted change -9.6%, CI: -17.4,-1.9; P = 0.015); however, neither baseline to endline change among controls (P = 0.926) nor the difference in change between groups (P = 0.090) were statistically significant.

Access to medication for adult chronic illness improved in both groups. The proportion of individuals reporting ever facing difficulties obtaining medication for their chronic illness decreased significantly in MPC recipients (adjusted change: -14.0%, CI: -21.9,-6.1; P = 0.001) and controls (adjusted change: -10.7%; CI: -20.1,-1.2; P = 0.028), though the difference between these changes was not statistically significant (P = 0.573). Medication affordability followed a similar trend with significant decreases in the proportion of individuals unable to afford medication for their chronic condition in the MPC (-20.7%; P>0.001) and control groups (-15.8%, P = 0.001) but with no significant difference in change between groups (P = 0.405).

## Health expenditures

Health expenditure for the most recent child, adult acute, and adult chronic illnesses (within the past six months), as well as overall household health expenditures were also evaluated. Baseline and endline descriptive analyses for each group are provided in S2 Table; individual change over time and differences in change between study groups are provided in Table 5; baseline and endline mean overall costs for child illness, adult acute illness, and overall household health expenditures are presented in Fig 3.

After adjusting for sociodemographic characteristics, change in nearly all health expenditure outcomes were not significantly different (P>0.05) between MPC recipients and controls. Despite this, change among MPC recipients and/or controls were significant for selected expenditure measures.

With regard to the most recent care visit for a child's illness, the proportion of controls with an out-of-pocket payment to the health facility increased 11.5% (CI: 4.6,18.3; P = 0.001) from baseline to endline; this also increased among MPC recipients, though not statistically significantly (P = 0.107). Consequently, facility payment amounts among controls were also 3.5 times higher (CI: 1.6,8.0; P = 0.001) at endline relative to baseline, though the analogous increase among MPC recipients was not statistically significant (P = 0.146). In adjusted models, the proportion of households with payment for medication obtained outside the facility significantly decreased 9.1% among controls (P = 0.049), but change among MPC households (P = 0.866) and the difference in change between groups were not significant (P = 0.267); no significant change or group difference in change was observed for medication costs (MPC P = 0.909; control P = 0.148; DiD P = 0.406). Overall payments for child illness significantly increased 9.1% (CI: 1.6,16.7%; P = 0.018) among controls, as did overall payment amounts, which were 2.8 (CI: 1.1,7.0; P = 0.033) times higher at endline than baseline; however, changes in both the proportion of households with payments and mean payment amounts were not significant among MPC recipients (P = 0.969 and P = 0.987, respectively) and did not significantly differ between groups (P = 0.159 and P = 0.214).

While health facility and overall expenditures for adult acute care did not significantly change in either group, the proportion of households with out-of-pocket payments for medication obtained at a pharmacy or elsewhere significantly decreased for both the MPC (-17.8%, CI: -30.9,-4.7; P = 0.008) and control (-19.0%, CI: -30.9,-7.0; P = 0.002) groups. Accordingly, adjusted medication costs also decreased similarly (DiD P = 0.895) for both groups; MPC group medication costs at endline were 0.2 (CI: 0.0,0.7; P = 0.016) times those at baseline and control costs at endline were 0.1 (CI: 0.0,0.5; P = 0.003) times baseline amounts. MPC recipients saw more significant changes in medication expenditures for adult chronic illness relative to controls. Average monthly medication costs were reported by significantly fewer individuals in both groups at endline compared to baseline with an adjusted change of -14.0% (CI: -21.5,-6.5%; P<0.001) among MPC recipients and -10.0% (CI: -18.6,-1.3; P = 0.024) among controls (DiD P = 0.471). Monthly medication payment amounts also significantly decreased for MPC recipients (0.2, CI: 0.0,0.7; P = 0.016) and controls (0.3, CI: 0.1,0.8; P = 0.019). Neither group saw significant change in the proportion with facility payments (MPC P = 0.055; control P = 0.840; DiD P = 0.245) or payment amounts (MPC P = 0.101; control P = 0.831; DiD P = 0.336) for the most recent chronic illness care visit.

Total household health expenditures in the past month were significantly higher among MPC households (Fig 3) and increased from baseline to endline in both groups in adjusted analyses. While change was not significant among MPC recipients (P = 0.194), expenditures were 2.8 (CI: 1.2,6.4; P = 0.016) times higher among controls at endline than at baseline. Change in routine health spending did not significantly differ between groups (adjusted DiD

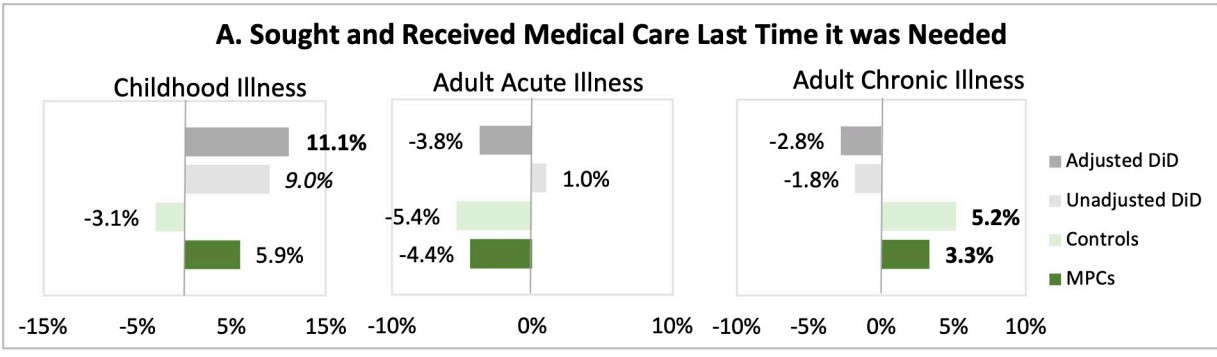

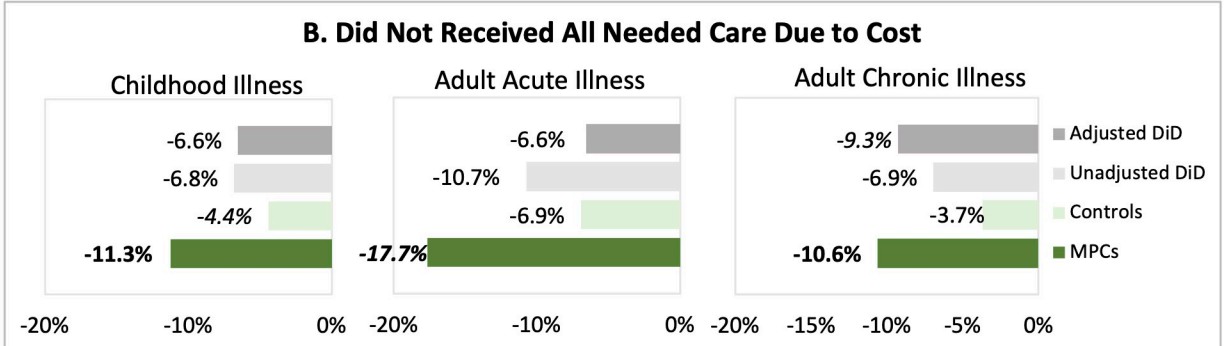

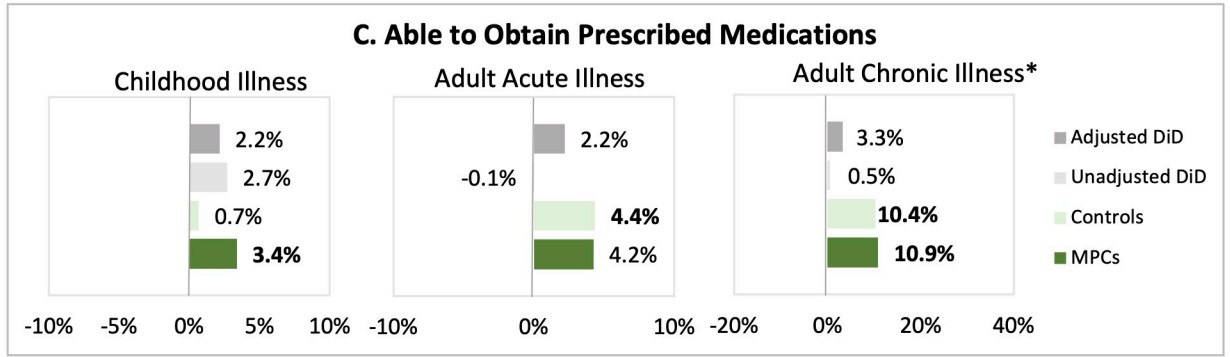

*Never faced difficulties obtaining medication

**Bold** indicates significant difference (either confidence internal does not overlap zero or P value < 0.05 for difference-in-difference comparison)

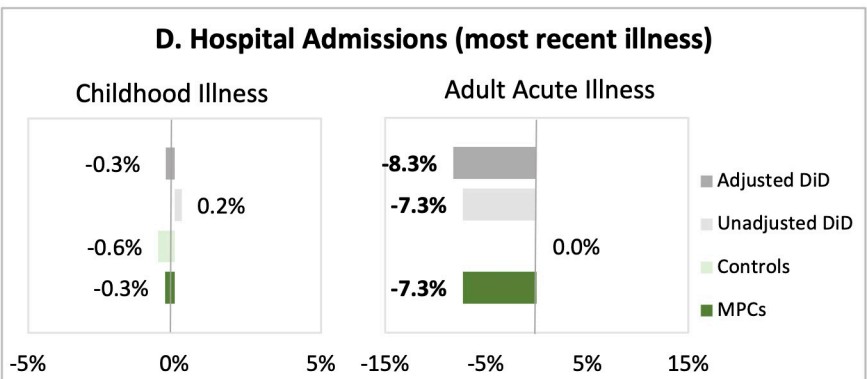

**Fig 2. Change in health utilization measures by group and difference in magnitude of change between groups.**

P = 0.405). Asset sales to pay for health expenses in the past three months were uncommon among participants (<11% in both groups). Although asset sales increased in both groups during the study period, change was only statistically significant for controls, among whom asset

**Table 5. Change in log health expenditures for most recent care and in the preceding month[a].**

| | UNADJUSTED CHANGE OVER TIME | | | | | | ADJUSTED CHANGE OVER TIME [b] | | | | | | | |
|---|---|---|---|---|---|---|---|---|---|---|---|---|---|---|
| | MPC HHs | | Control HHs | | DiD | | MPC HHs | | Control HHs | | DiD | | Effect Size | R² |
| | % | (95% CI) | % | (95% CI) | % | (95% CI) | % | (95% CI) | % | (95% CI) | % | (95% CI) | | |
| **Most Recent Childhood Illness** | | | | | | | | | | | | | | |
| **Health Facility Payments for OP Care [c]** | | | | | | | | | | | | | | |
| Any payment for outpatient care at facility | *9.3%* | *(-0.2,18.7)* | **11.2%** | **(4.9,17.5)** | -1.9% | (-13.3,9.4) | 8.2% | (-1.8,18.1) | **11.5%** | **(4.6,18.3)** | -3.3% | (-15.3,8.7) | -4.1% | 0.072 |
| Total paid at facility for visit (all HHs) | *1.0* | *(-0.1,2.2)* | **1.2** | **(0.4,1.9)** | -0.2 | (-1.5,1.2) | 0.9 | (-0.3,2.0) | **1.3** | **(0.5,2.1)** | -0.4 | (-1.8,1.0) | -68.6% | 0.072 |
| **Medication Costs at Pharmacy/ Elsewhere** | | | | | | | | | | | | | | |
| Any payment for medication outside facility | 0.3% | (-10.4,11.0) | -7.0% | (-15.5,1.5) | 7.3% | (-6.3,21.0) | -1.0% | (-12.7,10.7) | **-9.1%** | **(-18.2,0.0)** | 8.1% | (-6.2,22.4) | 22.1% | 0.044 |
| Total paid for medication (all HHs) | 0.0 | (-1.3,1.2) | -0.6 | (-1.6,0.4) | 0.6 | (-1.1,2.2) | -0.1 | (-1.5,1.3) | -0.8 | (-1.9,0.3) | 0.7 | (-1.0,2.4) | -14.1% | 0.044 |
| **Total Amount Paid for Illness [d]** | | | | | | | | | | | | | | |
| Any expense for most recent illness | 2.2% | (-8.0,12.3) | **9.1%** | **(2.1,16.2)** | -7.0% | (-19.3,5.4) | -0.2% | (-11.2,10.8) | **9.1%** | **(1.6,16.7)** | -9.4% | (-22.4,3.7) | -12.7% | 0.052 |
| Total cost for most recent illness (all HHs) | 0.3 | (-0.9,1.5) | **1.0** | **(0.2,1.9)** | -0.8 | (-2.3,0.7) | 0.0 | (-1.3,1.3) | **1.0** | **(0.1,1.9)** | -1.0 | (-2.6,0.6) | 5.2 | 0.053 |
| **Most Recent Acute Adult Illness** | | | | | | | | | | | | | | |
| **Health Facility Payments for OP Care [c]** | | | | | | | | | | | | | | |
| Any payment for outpatient care at facility | *11.0%* | *(-1.3,23.4)* | 7.8% | (-1.9,17.5) | 3.3% | (-12.4,18.9) | 9.2% | (-4.4,22.9) | 6.5% | (-3.7,16.6) | 2.8% | (-13.8,19.4) | 3.6% | 0.080 |
| Total paid at facility for visit (all HHs) | 0.9 | (-0.6,2.4) | 0.4 | (-0.7,1.6) | 0.5 | (-1.4,2.4) | 0.9 | (-0.8,2.5) | 0.2 | (-1.0,1.4) | 0.6 | (-1.3,2.6) | 170.4% | 0.085 |
| **Medication Costs at Pharmacy/ Elsewhere** | | | | | | | | | | | | | | |
| Any payment for medication outside facility | **-17.9%** | **(-29.3,-6.4)** | **-18.4%** | **(-29.6,-7.2)** | 0.6% | (-15.5,16.6) | **-17.8%** | **(-30.9,-4.7)** | **-19.0%** | **(-30.9,-7.0)** | 1.2% | (-16.2,18.6) | 4.6% | 0.086 |
| Total paid for medication (all HHs) | **-2.1** | **(-3.4,-0.7)** | **-2.1** | **(-3.5,-0.8)** | 0.1 | (-1.9,2.0) | **-1.9** | **(-3.4,-0.3)** | **-2.3** | **(-3.7,-0.8)** | 0.4 | (-1.7,2.5) | -5.9% | 0.087 |

*(Continued)*

**Table 5.** (Continued)

| | UNADJUSTED CHANGE OVER TIME | | | | | | ADJUSTED CHANGE OVER TIME [b] | | | | | | Effect Size | R$^2$ |
|---|---|---|---|---|---|---|---|---|---|---|---|---|---|---|
| | MPC HHs | | Control HHs | | DiD | | MPC HHs | | Control HHs | | DiD | | | |
| | % | (95% CI) | % | (95% CI) | % | (95% CI) | % | (95% CI) | % | (95% CI) | % | (95% CI) | | |
| **Total Amount Paid for Illness [d]** | | | | | | | | | | | | | | |
| Any expense for most recent illness | 1.9% | (-10.3,14.1) | 6.0% | (-4.1,16.1) | -4.1% | (-19.9,11.7) | -1.2% | (-15.1,12.7) | 7.9% | (-3.3,19.1) | -9.1% | (-26.7,8.5) | -12.4% | 0.069 |
| Total cost for most recent illness (all HHs) | 0.0 | (-1.5,1.5) | 0.4 | (-0.9,1.7) | -0.4 | (-2.3,1.6) | -0.3 | (-2.0,1.3) | 0.6 | (-0.8,2.1) | -0.9 | (-3.1,1.2) | 3.2 | 0.076 |
| **Most Recent Adult Chronic Illness Visit** | | | | | | | | | | | | | | |
| **Health Facility Payments for OP Care [c]** | | | | | | | | | | | | | | |
| Any payment for outpatient care at facility | 4.1% | (-2.6,10.8) | -0.9% | (-9.2,7.5) | 5.0% | (-5.8,15.7) | 7.8% | (-0.2,15.7) | 0.9% | (-8.0,9.8) | 6.8% | (-4.7,18.4) | 11.2% | 0.039 |
| Total paid at facility for visit (all HHs) | 0.3 | (-0.5,1.1) | -0.1 | (-1.1,1.0) | 0.4 | (-0.9,1.7) | 0.8 | (-0.2,1.8) | 0.1 | (-1.0,1.2) | 0.7 | (-0.7,2.1) | -46.7% | 0.047 |
| **Average Monthly Medication Costs** | | | | | | | | | | | | | | |
| Any regular (monthly) medication costs | *-13.6%* | *(-19.6,-7.6)* | -7.8% | (-16.1,0.5) | -5.8% | (-16.0,4.4) | *-14.0%* | *(-21.5,-6.5)* | -10.0% | (-18.6,-1.3) | -4.0% | (-15.1,7.0) | -6.8% | 0.061 |
| Average monthly medication costs (all cases) | *-1.7* | *(-2.5,-1.0)* | *-1.0* | *(-2.0,0.0)* | -0.7 | (-2.0,0.6) | *-1.8* | *(-2.7,-0.8)* | -1.3 | (-2.3,-0.2) | -0.5 | (-1.9,0.9) | 0.3 | 0.065 |
| **Routine Spending on Health** | | | | | | | | | | | | | | |
| Health expenditures (past month) [e] | 0.2 | (-0.5,0.9) | *0.7* | *(-0.1,1.5)* | -0.5 | (-1.6,0.5) | 0.5 | (-0.3,1.4) | **1.0** | **(0.2,1.9)** | -0.5 | (-1.6,0.6) | -88.9% | 0.095 |
| Sold assets to pay for health (past 3 months) | 0.8% | (-2.5,4.0) | 1.8% | (-1.8,5.4) | -1.0% | (-5.8,3.8) | 1.6% | (-2.4,5.6) | **3.8%** | **(0.0,7.5)** | -2.2% | (-7.4,3.0) | -24.9% | 0.022 |

*(Continued)*

**Table 5.** (*Continued*)

| | UNADJUSTED CHANGE OVER TIME | | | | | | ADJUSTED CHANGE OVER TIME [b] | | | | | | Effect Size | $R^2$ |
|---|---|---|---|---|---|---|---|---|---|---|---|---|---|---|
| | MPC HHs | | Control HHs | | DiD | | MPC HHs | | Control HHs | | DiD | | | |
| | % | (95% CI) | % | (95% CI) | % | (95% CI) | % | (95% CI) | % | (95% CI) | % | (95% CI) | | |
| Borrowed to pay for health (past 3 months) | -4.8% | (-11.0,1.4) | 2.1% | (-4.2,8.4) | -6.9% | (-15.8,1.9) | -7.7% | **(-14.8,-0.6)** | 2.5% | (-4.2,9.3) | **-10.3%** | **(-19.9,-0.6)** | -28.3% | 0.019 |

DiD = difference-in-difference; HH = household; OP = outpatient.

DiD represents the difference in change between MPC & control groups; effect size indicates the magnitude of this difference (DiD estimate divided by overall mean).

Bold italic indicates statistically significant (P < 0.001) findings; bold indicates statistically significant (P < 0.05) findings; italic indicates statistically significant (P < 0.10) findings.

[a] Analyzed as log US$; exchange rate: 1 JOD = 1.41 US$

[b] adjusted analyses controlled for principal applicant sex, age, education level, and marital status; household size and composition including the presence of household members in need of daily support, with chronic conditions, and children under five years old; dependency ratio; total household expenditure in the prior month; and receipt of humanitarian assistance, specifically total value of cash assistance received in the prior month, and current WFP beneficiary status/transfer modality (voucher or Choice)

[c] includes consultation fees, diagnostic testing and medications obtained at health facility during the initial visit to health facility, hospital outpatient department, or emergency room (without overnight stay)

[d] includes health facility payments for outpatient care and medications purchased at pharmacies outside health facilities (does not include referrals, in-patient care, and transportation)

[e] at facilities and for medication

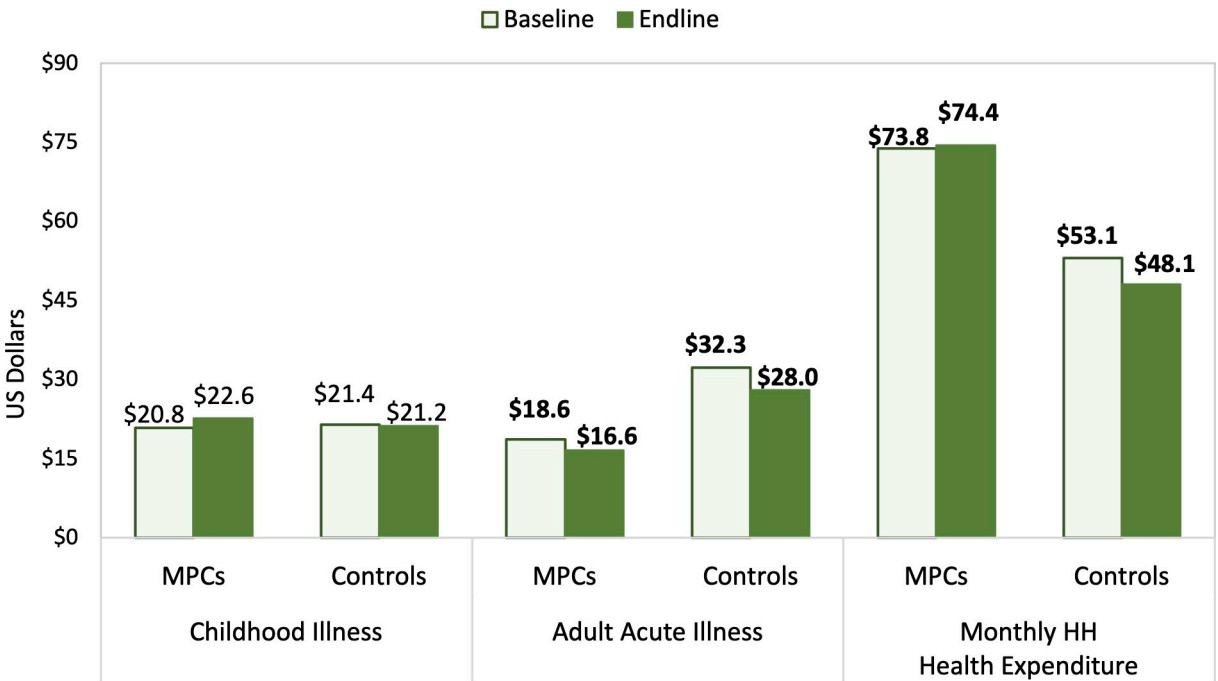

* **Bold** text indicates statistically significant difference between MPC/control groups at respective time period

**Fig 3. Health expenditures by group (one-year period from baseline to endline).**

sales increased 3.8% (CI: 0.0,7.5%; P = 0.049) (MPC adjusted P = 0.679); there was no significant difference in change between groups (adjusted P = 0.412). Borrowing money to pay for health expenses was more common in both groups, with marginally more MPC households borrowing money at baseline compared to controls (39.9% and 33.7%, respectively, P = 0.069), but similar proportions in both groups at endline (35.0% and 35.8%, respectively; P = 0.820). In adjusted models, borrowing decreased 7.7% (CI: -14.8,-0.6%; P = 0.034) among MPC households and increased 2.5% (CI: -4.2,9.3%; P = 0.461) among controls with a significant difference in change of -10.3% (CI:-19.9,-0.6; P = 0.037; effect size: -28.3%) between groups.

## Discussion

Frequency of needing, though not necessarily seeking, care for child and adult acute illness was similar among MPC recipients and controls at baseline; however, rates for actualized care-seeking for child and adult acute illness were significantly higher among control households than MPC households at baseline. Changes over the study period for care-seeking for both child and adult illness within groups were not statistically significant; however, the between group difference in change for child care-seeking was statistically significant, which is reflective of increases in care-seeking in the MPC group and decreases in the control group. Across all types of care, cost was the predominate barrier for those not seeking care and among care recipients, many did not receive all recommended care due to cost. The proportion of care-seekers unable to afford all recommend care was statistically similar between the two comparison groups at baseline for child illness, adult acute illness, and chronic illness. At the end of the one year follow up period, statistically significant decreases in the proportion of households not receiving all needed care due to cost were reported for all three illness categories in the MPC group, suggesting that MPC may improve the affordability of care. Similar statistically significant declines were not observed among control households and there were also no significant differences in change over the study period between the two groups for two of the three categories of illness (a difference was observed for chronic illness), yielding a mixed result with respect to MPC's impact on the ability to afford all needed care. While data from 2019 is not yet available, UNHCR's annual Health Access and Utilization Survey (HAUS) demonstrates similarly mixed care-seeking trends over time. Although the proportion of households surveyed for the HAUS that sought care in the preceding month decreased substantially from 84% in 2016 to 45% in 2018, access to medical services for chronic conditions improved over the same period and perceived ability to afford necessary medical services for chronic conditions also greatly improved from 2017 to 2018 [5–7]. Mixed findings on the impact of MPC on care-seeking in the present study are aligned with increased child health care-seeking observed in a recent study of MPC in Lebanon, but in contrast to null findings of two MPC evaluations in Jordan [31–33].

With respect to access to medicines, more than 95% of households in both groups were able to afford prescribed medications for child and adult acute illness at baseline possibly because many of these medications are included in the essential care package. MPC receipt was associated with marginally significant increases in access to medicines for both child illness and adult acute illness. Cost was a barrier to medication for a larger proportion of households that had a member with a chronic health condition at baseline than at endline with similar levels of change in both control and MPC households. Increased affordability of chronic disease medications was not related to MPC receipt; it is possible but unlikely that this is attributable to the March 2019 policy changes intended to reduce out-of-pocket expenditures because endline data collection was conducted soon after the change and many households would not yet have benefited from it. Improvements in access to medication for chronic conditions observed in

our study contrast with a decline from 2016 through 2018 previously noted in UNHCR's HAUS [5–7]. The HAUS did, however, note improvements in availability of chronic illness medications from 2017 to 2018, which, coupled with policy changes and care-seeking behaviors, may have implications for medication affordability.

Changes in the sector where care was received were assessed for child illness and adult acute and chronic illness, but were not significant for either group with the exception of a decline in public sector utilization for chronic illness among controls, suggesting that increased use of charity/non-governmental organization (NGO) facilities was not a factor in increasing affordability of chronic disease medication and that MPC was not associated with a shift in care-seeking away from the public and charity/NGO sectors to the private sector. This is a positive finding with respect to the objective of providing unconditional cash to meet basic needs and indicates that MPC receipt does not result in replacement of public sector care with more costly care in the private sector. UNHCR's HAUS reported a contradictory shift from comparable utilization of government hospitals and private clinics in 2016 and 2017 to markedly higher refugee use of private pharmacies for care-seeking in 2018, likely due to GoJ policy changes increasing the cost of public sector care in early 2018 [5–7]. The absence of significant changes in our findings and, although not significant, decreased private sector utilization for most illness types may support the potential influence of 2019 policy changes decreasing public sector care costs.

Emergency room visits for child illness and adult acute illness declined over the study period for both MPC recipients and controls. For children, declines were statistically significant for both MPC and control households, but the magnitude of decline was marginally greater among MPC households; among adults a statistically significant decline was observed only in the MPC group and the difference in change between groups over the study period was not significant. The observed decreases in emergency room use are in contrast with what would be expected as a result of April 2019 GoJ policy changes, which should have translated to decreased out of pocket payments; however, it is likely that many refugees were not widely aware of policy changes (in particular if they had not sought care recently) or that changes were not fully implemented prior to data collection [4]. Hospitalizations remained constant for children but statistically significantly declined for adult acute illness among MPC recipients; the difference in change over the study period was statistically significant, suggesting that MPC receipt was associated with a decrease in hospital admissions for adult acute illness. Collectively, this suggests that MPC may facilitate better access to care and prevent hospitalizations, which are important impacts for individuals and for financing of the broader humanitarian health response where reduced hospitalizations translates to a significant cost savings. In the parallel study conducted in Lebanon, results were also mixed with MPC recipients having a significantly smaller increase in child hospitalization compared to controls (DiD -6.1%; P = 0.043) yet no significant differences for adult hospitalizations [26].

Regarding health expenditures, the proportion of MPC recipients that reported out of pocket expenses and average [log transformed] expenditures were assessed for child illness and adult acute and chronic illness. Total child health expenditures at the most recent visit were constant between groups and over time; adult health expenditures were significantly higher among controls at both baseline and endline and in both groups. Total expenditures for the most recent adult acute illness were more varied; controls had consistently higher expenditures and both groups saw a decrease in expenditures over the study period. However, when analyzed as a proportion of households with payments and log transformed expenses, there were no significant differences in change in expenditures between the two groups for any health expenditure measure, demonstrating that MPC receipt was not associated with increased spending on health during the study period. Examination of total monthly

household health expenditures reveals significantly higher expenditures among MPC recipients compared to controls at both baseline and endline. This aligns with evidence from Lebanon including our parallel study, which also observed significantly greater monthly household health expenditures among MPC recipients at both baseline and endline [26, 31]. There was a significant reduction in the proportion of households in the MPC group borrowing to pay for health expenses and when change over time was compared between the two groups, MPC receipt was associated with a reduction in borrowing to pay for health, suggesting that MPC was protective for household financial risks associated with health, though this same trend was not observed for asset sales to pay for health costs, which were less frequently reported. Overall reduced health expenditures in our findings align with temporal trends reported in the HAUS where household health expenditures decreased annually from 2016 through 2018 [5–7]. Our findings also echo those from two previous evaluations of cash assistance for Syrian refugees in Jordan, which both found no statistically significant association between MPC receipt and health expenditures [32, 33]. It is worth noting that while a 2017 study comparing CVA provided by UNHCR, UNICEF, and WFP in Jordan found no evidence of increased health spending, they did find that beneficiaries believed receiving cash assistance, and in turn having more fungible income, had improved their ability to seek needed health care [32]. Previous studies of Syrian refugee MPC assistance throughout the region, notably in Lebanon and Turkey, also found limited or no impact of MPC on health expenditures, suggesting the probable influence of policy changes and other factors on health spending [31, 34].

Average monthly household expenditures exceeded the Minimum Expenditure Basket (MEB), which reflects the monthly expenditure amount needed to meet basic needs and is typically calculated on a per capita or household size basis, in both the MPC and control groups at both baseline and endline [35]. MPC assistance amounts were considerable and equate to approximately 46–51% of average monthly expenditures. Monthly household health expenditures among MPC recipient households averaged US$20.6 (baseline) to US$26.3 (endline) more than control households, which translates to a difference in household health expenditures of US$247.2 to US$315.6 annually. Applying these figures to the approximately 28,000 Syrian refugee households that received UNHCR MPC from mid-2018 to mid-2019, it can be estimated that MPC expenditures contribute US$6.9–8.8 million annually to refugee health in Jordan, equating to approximately 10% of health sector response funding, which was US$73 million in 2019 [36]. Although MPC assistance amounts were sizable in this context, post distribution monitoring during this time found that cash assistance was more often spent on rent, utilities, and food (though still frequently used on health), suggesting that higher transfer amounts may be needed before households increase spending on health [37]. While MPC increased spending on health and affordability of care in this study, they cannot be viewed as a replacement for direct support to the health sector and they should be considered as one of several options for increasing access to care. There are few comparative studies of MPC compared to conditional cash transfers, which either have qualifying criteria (e.g., for individuals with chronic health conditions) or use conditions (e.g., ability to demonstrate care-seeking or medication purchase); however, previous evidence from development settings has demonstrated the positive impact of conditional cash transfers on health behaviors and, to a lesser extent, health outcomes [38–40]. Much of this research is limited to Latin-American countries and does not capture the nuanced interplay of factors in displacement settings, hindering the ability to translate inferences to humanitarian contexts in other regions [38–40]. Moreover, the potential benefits of cash transfers on improved health outcomes are noted to rely on the provision of quality care rather than utilization alone, supporting the need to integrate supply-side interventions when implementing cash assistance to improve health [40].

While affordability is constantly a primary barrier to healthcare access and utilization in Jordan and other humanitarian contexts, competing demands on households, among other factors, often influence health-related behaviors, likely hindering the impact of intervention via MPC alone. Several reviews of cash transfers in development settings have emphasized the influence of non-financial factors on health in explaining areas of heterogeneity in the impacts of cash, furthering the idea that cash alone, whether conditional or unconditional, is likely most effective when combined with supply-size or health system strengthening interventions [38–40]. Moreover, evidence of the benefits of health education and community-based screening, monitoring, and counseling interventions for improving chronic disease care in humanitarian settings is increasing, but alone these interventions cannot address affordability barriers [41–44].

Previous evidence from Syrian refugees in Jordan suggests that conditional cash transfers and health education may be more effective in improving health indicators for chronic disease compared to MPC [45]. Health sector 'top ups,' where MPC transfer value is increased for households with specific characteristics (e.g., members with a chronic health condition or disability) necessitating higher routine health expenditures could be another strategy to deliver cash for health at scale, though this approach has not been tested in a humanitarian setting to our knowledge.

## Limitations

This study had several limitations. First, results may be limited by the quality of expenditure reporting given enumerator and respondent confusion about what to include in the various expenditure questions despite thorough training on interview techniques and probing for accurate responses. Self-report of expenditure data may also have introduced recall bias, particularly for households whose most recent care-seeking was months prior to interview. Second, expansion of WFP's Choice program during the study period resulted in approximately half of participants in both the MPC and control groups switching to WFP Choice. Because the Choice program allows beneficiaries to receive unconditional cash assistance, although this was accounted for in adjusted analyses, the influence of WFP transfer modality changes on our outcomes cannot be precluded. Changes to the GoJ health policy directing healthcare costs for Syrian refugees near the beginning and end of the study may likely have influenced care utilization and are potential confounder. The analysis approach assumes that both comparison groups would have parallel trends during the study period if no intervention was received; however, it is difficult to determine the credibility of this assumption because pre-baseline data that could be used to assess trends prior to the study period are not available. In addition, the study assumes that both comparison groups are similarly impacted by GoJ policy changes and other temporal trends due to their similarity as vulnerable refugees residing in the same location, however, it is possible that they experienced differential impacts, which could have contributed to [lack of] differences in change over time that are attributed to the intervention. The timing of this change also adds difficulty to interpreting changes in health expenditures at public sector facilities. Small sample sizes should be considered in interpretation of findings. While sample sizes for those not seeking care across care categories (i.e., child, adult acute, and adult chronic illness) prohibited robust analysis of change, conditional denominators for many outcomes (e.g., those who needed care, sought care, etc.) restricted the analyzed sample well below the minimum sample sizes calculated in the initial study design. This likely affected the sufficiency of statistical power to detect significant differences between groups. Care should similarly be taken in interpreting DiD estimates in cases where comparison groups significantly change, though in different directions. Finally, ideally this study would be conducted with new rather than existing beneficiaries but, given the extended nature of the refugee crisis and the need to conduct research within the ongoing humanitarian response, such a design was not possible.

## Conclusions

The impacts of unrestricted cash transfers, which provided for approximately half monthly household expenditures, on Syrian refugee health outcomes in Jordan were varied. There were no significant changes in household health expenditures during the study period, though cash transfer recipients were significantly less likely to report borrowing to pay for health expenses. At both study time points significantly higher household health expenditures were found among cash recipients, which may indicate that benefits are realized closer to transfer initiation and then sustained. Improvements in care-seeking attributed to receipt of unrestricted cash transfers were observed for child illness but not acute or chronic illness among adults. The proportion of households unable to receive all needed care due to cost declined significantly in the MPC group for child illness and both acute and chronic illness among adults, suggesting that cash may have improved health access in at least some way, though differences in change compared to the control group were not significant. Medication was accessible for nearly all households for both child illness and adult acute illness, but not chronic disease; access and affordability of chronic disease medications improved over the study period for both groups and not as a result of cash transfers. Cash transfers were not associated with a shift in care-seeking away from the public and charity/NGO sectors to the private sector and appear to yield benefits in terms of reduced hospitalizations for adult acute illness. These are both positive findings with respect to humanitarian response financing, particularly because investment in unrestricted cash transfers may translate to savings in the health sector response.

## Supporting information

**S1 Checklist. Inclusivity in global research.** A complete copy of PLOS' questionnaire on inclusivity in global research.
(DOCX)

**S1 File. Supplemental methods.** Additional details on sampling methods, including change in intervention receipt among study households during the study period and analyzed sample follow-up by intervention receipt.
(PDF)

**S2 File. Multi-purpose cash transfers and health among vulnerable Syrian refugees in Jordan: Study questionnaire.** The questionnaire developed and used for this study.
(PDF)

**S1 Table. Health care-seeking and medicines for household member illness at baseline and endline.** Baseline and endline descriptive analyses of care-seeking outcomes by group.
(PDF)

**S2 Table. Health expenditures for most recent child, adult acute, and adult chronic illness care and in the preceding month (USD) at baseline and endline.** Baseline and endline descriptive analyses of health expenditure outcomes by group.
(PDF)

## Acknowledgments

We would like to thank Elsa Groenveld and Emily Chambers Sharpe from Medair for their support in study conceptualization and planning; Harry Brown from UNHCR for assistance with data sharing and sample planning; and Kayla Pfeiffer-Mundt from JHSPH for support

with data cleaning. We are also grateful to Eman Saleh and the Medair interviewer team for their efforts during data collection.

## Author Contributions

**Conceptualization:** Shannon Doocy.

**Formal analysis:** Emily Lyles, Antonio Trujillo.

**Funding acquisition:** Shannon Doocy.

**Methodology:** Paul Spiegel, Ann Burton, Shannon Doocy.

**Project administration:** Emily Lyles, Stephen Chua, Yasmeen Barham, Dina Jardenah.

**Supervision:** Emily Lyles, Shannon Doocy.

**Writing – original draft:** Emily Lyles, Shannon Doocy.

**Writing – review & editing:** Emily Lyles, Stephen Chua, Yasmeen Barham, Dina Jardenah, Antonio Trujillo, Paul Spiegel, Ann Burton, Shannon Doocy.

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
