## [Decision Letter · Decision Letter 0]

31 Mar 2022

PGPH-D-21-01130

Multi-purpose cash transfers and health among vulnerable Syrian refugees in Jordan: A prospective cohort study

Dear Dr. Doocy,

Thank you for submitting your manuscript to PLOS Global Public Health. After careful consideration, we feel that it has merit but does not fully meet PLOS Global Public Health’s publication criteria as it currently stands. Therefore, we invite you to submit a revised version of the manuscript that addresses the points raised during the review process.

We look forward to receiving your revised manuscript.

Kind regards,

Stefan Kohler

Academic Editor

Journal Requirements:

2. Please include a complete copy of PLOS’ questionnaire on inclusivity in global research in your revised manuscript. Our policy for research in this area aims to improve transparency in the reporting of research performed outside of researchers’ own country or community. The policy applies to researchers who have travelled to a different country to conduct research, research with Indigenous populations or their lands, and research on cultural artefacts. The questionnaire can also be requested at the journal’s discretion for any other submissions, even if these conditions are not met.  Please find more information on the policy and a link to download a blank copy of the questionnaire here: https://journals.plos.org/globalpublichealth/s/best-practices-in-research-reporting. Please upload a completed version of your questionnaire as Supporting Information when you resubmit your manuscript.

Additional Editor Comments (if provided):

Reviewers' comments:

Reviewer's Responses to Questions

**Comments to the Author**

1. Does this manuscript meet PLOS Global Public Health’s publication criteria? Is the manuscript technically sound, and do the data support the conclusions? The manuscript must describe methodologically and ethically rigorous research with conclusions that are appropriately drawn based on the data presented.

Reviewer #1: Partly

Reviewer #2: Partly

2. Has the statistical analysis been performed appropriately and rigorously?

Reviewer #1: Yes

Reviewer #2: Yes

3. Have the authors made all data underlying the findings in their manuscript fully available (please refer to the Data Availability Statement at the start of the manuscript PDF file)?

Reviewer #1: Yes

Reviewer #2: Yes

4. Is the manuscript presented in an intelligible fashion and written in standard English?

Reviewer #1: Yes

Reviewer #2: Yes

5. Review Comments to the Author

Reviewer #1: The paper provides evidence from the MPC in Jordan and its relation with the health care need, utilization and expenditure of refugees. The study's main strength is the data collection that addresses to the aim of the study. Even if I find the findings of the study very interesting I have some reservations about the institutional set up during the study period. Especially potential uncertainty generated by the institutional set up which makes it difficult to interpret the analysis results. My main concerns are around i) GOJ's health policy change in March 2018 and March 2019, ii) UNHCR's revision of the MPC eligibility criteria (lines 179-180) and resulting change in the eligibility status of some families, iii) WFP's coverage change during the study period. The authors do explain these limitations thoroughly in the Discussion session. I believe the study will improve if these are brought up earlier in the Introduction and the institutional set up and their relation to the study findings will be better understood by the readers. Especially, the authors should put more emphasis on whether we should expect any differential effect of these changes on the treatment and control groups.

Additionally here are couple of points that needs clarification:

Table 2: The treated move to apartments by the endline. Why would that happen? Are there any concurrent policies about the housing of the refugees? Does this have to do with the MPC? How does this affect the refugees health? Do they also change neighborhood?

Table S3:I see that the outpatient visits increase for control and treatment by the endline? This may be due to the GOJ policy change in March 2019 or a simultaneous increase in the health care workers supply or health infrastructure buildup. Needs to be elaborated.

Study Sample: What are the characteristics of the households that dropped from the program? These should be provided in a summary table.

MPC program eligibility: How do the authors interpret revisions in the MPC and their relation to the outcome variables? Do the program rules generate any incentives for the manipulation of health status or household composition for the applicants or beneficiaries?

Method: The inability to check the parallel trends assumption is a big drawback in this setting. There is reason to believe that the MPC recipients and the control follow different trends. One potential area of concern is the duration of cash transfers and how long the MPC recipients were supported or whether the control group households were previously supported or not. For this reason if the dataset includes a variable to control for these points, it will be nice to introduce it in the text.

Minor comments: the references need to be put in order. Reference 32 (mentioned in line 558) is not a study about Turkey.

Reviewer #2: This is an interesting study addressing the relevant topic of health financing in humanitarian settings and contributing to generating evidence on the modality of multi-cash vouchers. While this study clearly adds to the evidence base of this topic, it can benefit from a review to improve the quality of the manuscript especially in terms of results' structure and discussion. Below are some points to be addressed by authors.

Introduction: clear introduction and good framing of the study background especially in relation with the Jordanian context and introducing cash assistance. However, the comparison between cash assistance and in-kind assistance (of households/individuals?) might not apply to health as it does to other sectors. Authors can compare this modality to in-kind assistance of health centres or other health financing approaches and/or highlight the gaps in the literature about this comparison (here or in the discussion).

Methods:

- the 'risk of cross-over' justifying the selection of the target of population is not clear to readers and a reflection on this decision on the discussion could help generate inferences about the impact of MPC in other households.

- the selection of the most conservative proportion of 50% as well as the minimum difference level in the sample size calculation needs to be justified in relation to what expected based on similar study (Lebanon's study or perhaps (in the time of study design) other studies).

- table 1 (or at least the section on the analyzed sample) could fit better in the results section as it might lead to confusion among readers in its current position.

- please add an explicit list of eligibility criteria and consequently explain whether the adult household member (replacing the eligible participant) meets those criteria. A short description of the percentage of those participants and their characteristics is essential if available.

- please clarify the ethical arrangements to use the phone numbers of UNHCR-registered refugees and how the team handled the use of these identifiers.

Results:

- this is a very rich section, but could benefit from sub-headings to support the reader throughout the data presentation. Given the lack of variable definition section in the methods, some variables remain unclear (e.g. how is 'household size' is defined especially for across different residence type; 'in-kind assistance', etc).

- there is a discrepancy between the findings about household economy in table 3 and those in the corresponding text. please review them.

- please briefly explain the reasons behind the selection of confounders for the adjusted estimates, ie in the statistical analysis (either here or in the methods section)

- please explain why the visits to pharmacies were omitted from the sources of healthcare and how this major gap in the available data is addressed

- in the presentation and interpretation of findings, could you comment whether a statistically significant DiD could have been reached in case of higher sample size (ie increased power of the study), and whether inferences can (or not) be made especially when the adjusted estimates for the comparison groups are in different direction and statistically significant for instance.

- please avoid statements like 'not statistically smaller' (better use simply: not statistically different?) and 'somewhat' protective in describing the evidence emerging from the study.

Discussion:

It is recommended to write a concise discussion with less results (ie avoiding the repetition of detailed results and perhaps summary statements of the key findings for each outcome) and more comparison with the wider literature as well as the global health implications of the study and perhaps future research. here are some specific points to addressed :

- the first sentence is not clear - could you please explain the difference between frequency of needing care and care-seeking rates. please avoid introducing new variables and results in the discussion.

- In case of re-writing of the discussion section based on the previous recommendation, please use the space to discuss (with more details using a comparative lens for the current study) the mixed results from other studies in Jordan and Lebanon)

- the following statement "This may suggest that MPC effects on increased health expenditure were realized closer to when transfers were first received and then were sustained (and thus did not change during the study period)" is unclear and introduced confusion as transfer timing was not discussed before in the manuscript. Please explain or drop.

- the discussion referred to very interesting evidence from the literature about how to improve the use of cash assistance, but very briefly. Examples include the need to have higher transfer amounts (line 573) and the limited positive impact of conditional transfers on helath outcomes (line 581). Please elaborate more your discussion (with perhaps a quick reflection on the quality of evidence / studies especially in the case of the second example).

6. PLOS authors have the option to publish the peer review history of their article (what does this mean?). If published, this will include your full peer review and any attached files.

**Do you want your identity to be public for this peer review?** For information about this choice, including consent withdrawal, please see our Privacy Policy.

Reviewer #1: No

Reviewer #2: **Yes: **Ibrahim Bou-Orm

---

## [Decision Letter · Decision Letter 1]

12 Aug 2022

PGPH-D-21-01130R1

Multi-purpose cash transfers and health among vulnerable Syrian refugees in Jordan: A prospective cohort study

Dear Dr. Doocy,

Thank you for submitting your manuscript to PLOS Global Public Health. After careful consideration, we feel that it has merit but does not fully meet PLOS Global Public Health’s publication criteria as it currently stands. Therefore, we invite you to submit a revised version of the manuscript that addresses the points raised during the review process.

We look forward to receiving your revised manuscript.

Kind regards,

Stefan Kohler, Ph.D., M.D.

Academic Editor

Journal Requirements:

1. Please ensure that the funders and grant numbers match between the Financial Disclosure field and the Funding Information tab in your submission form. Note that the funders must be provided in the same order in both places as well.

Additional Editor Comments (if provided):

- Please revisit reviewer #1's comments and the related revisions and response.

- Please elarobare in the introduction how the submitted study adds to your related works [26] mentioned by reviewer #1.

Reviewers' comments:

Reviewer's Responses to Questions

**Comments to the Author**

1. If the authors have adequately addressed your comments raised in a previous round of review and you feel that this manuscript is now acceptable for publication, you may indicate that here to bypass the “Comments to the Author” section, enter your conflict of interest statement in the “Confidential to Editor” section, and submit your "Accept" recommendation.

Reviewer #1: No

Reviewer #2: All comments have been addressed

2. Does this manuscript meet PLOS Global Public Health’s publication criteria? Is the manuscript technically sound, and do the data support the conclusions? The manuscript must describe methodologically and ethically rigorous research with conclusions that are appropriately drawn based on the data presented.

Reviewer #1: Partly

Reviewer #2: (No Response)

3. Has the statistical analysis been performed appropriately and rigorously?

Reviewer #1: Yes

Reviewer #2: (No Response)

4. Have the authors made all data underlying the findings in their manuscript fully available (please refer to the Data Availability Statement at the start of the manuscript PDF file)?

Reviewer #1: Yes

Reviewer #2: (No Response)

5. Is the manuscript presented in an intelligible fashion and written in standard English?

Reviewer #1: Yes

Reviewer #2: (No Response)

6. Review Comments to the Author

Reviewer #1: The authors addressed the limitations raised by the reviewers in the Letter. However, I do not see that these changes were made in the main text. In the Letter, the authors mention that "The introduction was revised to make more explicit the three main concerns noted by reviewer #1 earlier in the article." however I do not see these revisions taking place. In this way, the revised manuscript does not address to my main concerns regarding the methodology which leaves very limited scope to improve/distinguish this current work in comparison to what the authors already presented in a previous work, cited below.

Lyles E, Arhem J, Khoury G, Trujillo A, Spiegel P, Burton A, et al. Multi-purpose cash transfers and health among vulnerable Syrian refugees in Lebanon: A prospective cohort study. BMC Public Health. 2021;21: 1176. doi: 10.1186/s12889-021-11196-8.

Reviewer #2: (No Response)

7. PLOS authors have the option to publish the peer review history of their article (what does this mean?). If published, this will include your full peer review and any attached files.

**Do you want your identity to be public for this peer review?** For information about this choice, including consent withdrawal, please see our Privacy Policy.

Reviewer #1: No

Reviewer #2: **Yes: **Ibrahim Bou-Orm

---

## [Editor Report · Decision Letter 2]

10 Oct 2022

Multi-purpose cash transfers and health among vulnerable Syrian refugees in Jordan: A prospective cohort study

PGPH-D-21-01130R2

Dear Dr Doocy,

We are pleased to inform you that your manuscript 'Multi-purpose cash transfers and health among vulnerable Syrian refugees in Jordan: A prospective cohort study' has been provisionally accepted for publication in PLOS Global Public Health.

Best regards,

Stefan Kohler, Ph.D., M.D.

Academic Editor